# An Acceleration Based Fusion of Multiple Spatiotemporal Networks for Gait Phase Detection

**DOI:** 10.3390/ijerph17165633

**Published:** 2020-08-05

**Authors:** Tao Zhen, Lei Yan, Jian-lei Kong

**Affiliations:** 1College of Engineering, Beijing Forestry University, Beijing 100083, China; zhentao@bjfu.edu.cn; 2Artificial Intelligence Academy, Beijing Technology and Business University, Beijing 100048, China; 3National Key Laboratory of Environmental Protection Food Chain Pollution Prevention, Beijing 100048, China

**Keywords:** gait-phase-recognition, FMS-Net, spatiotemporal networks, IMU signals, skip-connection structure

## Abstract

Human-gait-phase-recognition is an important technology in the field of exoskeleton robot control and medical rehabilitation. Inertial sensors with accelerometers and gyroscopes are easy to wear, inexpensive and have great potential for analyzing gait dynamics. However, current deep-learning methods extract spatial and temporal features in isolation—while ignoring the inherent correlation in high-dimensional spaces—which limits the accuracy of a single model. This paper proposes an effective hybrid deep-learning framework based on the fusion of multiple spatiotemporal networks (FMS-Net), which is used to detect asynchronous phases from IMU signals. More specifically, it first uses a gait-information acquisition system to collect IMU sensor data fixed on the lower leg. Through data preprocessing, the framework constructs a spatial feature extractor with CNN module and a temporal feature extractor, combined with LSTM module. Finally, a skip-connection structure and the two-layer fully connected layer fusion module are used to achieve the final gait recognition. Experimental results show that this method has better identification accuracy than other comparative methods with the macro-F1 reaching 96.7%.

## 1. Introduction

In recent years, robotic exoskeleton has become an emerging technology in medical, living, industrial and military applications. Among them, lower extremity exoskeleton has important research value in the medical field, its main potential is to enhance the patient’s ability to move in rehabilitation therapy, and enhance physical function after receiving treatment, and hope to improve their quality of life as much as possible. Among them, gait recognition technology is an important technical guarantee for the robot to process a large amount of instantaneous time series data, which is one of the most important features to display the posture and phase of each specific patient [1]. Therefore, there is an urgent need to accurately judge the gait phase of the human lower extremity state change in order to enhance the consistency and coordination of human-computer interaction [2]. In medical disease-diagnosis and rehabilitation research, effective analysis of gait phases has also achieved remarkable results, which has been used in clinical treatment plans for stroke, Parkinson’s disease, brain trauma and other diseases [3,4]. Note that traditional walking analysis is expressed by detecting different gait phases based on motion information (e.g., angle, speed or acceleration) of knees, ankles and hips while walking or running. For example, Fino et al. [5] used abnormal gait phases to detect concussion or mild head injury. Mathieu et al. [6] proposed a novel adaptive dynamic time warping based on Hidden Markov to analyze gait for identifying persons with physical disabilities and provide them with appropriate alerts by monitoring walking. In order to overcome the problem of poor adaptive ability caused by pure mechanical structure, some researchers have begun to recognize the phase of the lower limbs of the human body through programming and algorithms to achieve the purpose of controlling wearable auxiliary devices [7]. For instance, Ruiming [8] mentioned gait subphase recognition of high-quality is significant to the control of lower-limb powered exoskeletons.

According to previous studies, gait-phase-recognition methods can generally be divided into two categories. The first type is the threshold method, which determines the corresponding phase information by setting the corresponding threshold [1]. However, this type of algorithm is too rough and difficult to deal with complicated situations. In recent years, with the development of artificial intelligence technology, many researchers began to input different types of sensor data into deep learning models to achieve the purpose of detecting gait phase. For example, Mukherjee, etc. [9] proposed a deep-learning method using machine vision to detect pedestrian gait phase in real time. However, the camera is susceptible to interference from the external environment when capturing images. Moreover, this method is susceptible to the limitation of the use space. Ryu et al. [10] proposed an SVM method to process the electromyography (EMG) data collected during gait to identify the four sub-gait phases of pedestrians, thereby improving the above-mentioned problem of being easily interfered by the environment. However, commercial sEMG acquisition equipment is bulky, expensive and extremely inconvenient to wear, and it is easily affected by sweat stains during the collection of EMG data. In order to overcome the shortcomings of the above-mentioned technology, Ding et al. [11] proposed using the proportion-based fuzzy algorithm to process foot pressure signals to realize gait-phase-recognition, but the plantar pressure is also more susceptible to the wearer’s weight, shoe size and load. What is worse, the failure rate of pressure sensors is also relatively high, making it difficult to be widely applied in reality.

In recent years, many researchers have begun to study the use of inertial sensors (IMU) to achieve gait-phase-recognition methods. This is mainly because more abundant information of human movement can be obtained by using a small number of IMUs. Moreover, the IMUs are non-invasively installed on relevant parts of the body, which will not cause harm and too much inconvenience to the wearer [12]. Simultaneously—in the process of collecting IMU information—the signal is difficult to be interfered by the wearer’s own weight, load, and sweat on the wearing part. Compared with the detection method of plantar pressure or muscle electrical signal, the IMU detection method has obvious advantages. In addition, the cost of inertial sensors is relatively low [13], and the inertial information of certain parts of the body during the movement of the human body also has periodic characteristics. Many researchers place the IMU on the instep, calf and thigh. For example, Yan et al. [1] designed a voting-weighted integrated deep learning algorithm, and by inputting acceleration signals on the instep, calf and thigh into the model, successfully detected four subphases of pedestrians, and achieved very good results. Identify the effect. Zhen et al. [14] combined LSTM and DNN models to design an LSTM–DNN deep learning algorithm, and also detected four sub-phases of pedestrians by accelerating the acceleration signals on the instep, calf and thigh. Of course, using more IMUs can collect more phase information, but the requirements on the equipment will be very high, and more input signals will also cause a greater calculation burden on the recognition algorithm model. In addition, too many sensors will have a greater impact on the wearer. If there are multiple external sensor signal inputs, the model should first process the signals simultaneously before processing to ensure that the input signals are sent at the same time, which is undoubtedly very difficult for researchers. Therefore, the researchers gradually shifted their attention to the scheme of using a single IMU to detect the gait phase. For example, Manchola et al. [15] positioned the inertial sensor on the instep, used hidden Markov model as the phase recognition model, and obtained better results than the threshold-based algorithm. In view of the previous research, this article considers placing the IMU in the position of the lower leg, through continuous optimization algorithm model to obtain excellent recognition effect. Gohar et al. [16] Using the inertial information of the chest to realize the identification of personnel. It is important that this sensor application has little impact on personal activities and daily life, and is easy to wear, which is helpful for clinicians to judge early intervention, treatment effects and patient rehabilitation progress.

Although we have always been interested in IMU technology, there is still a lack of reality for actual research. This study uses multiple wearable sensors for verification to monitor walking status and gait stability in actual clinical practice. In a multi-IMU-based method, a sufficiently intelligent and diverse gait phase pattern recognition model with multi-sensor data preprocessing analysis and information fusion technology is the most critical problem to be solved for gait of the lower limbs to detect various wearable motion systems or rehabilitation Exoskeleton. To achieve this goal, some statistical learning or machine learning methods are used to calculate the spatiotemporal and biomechanical parameters of walking gait. The mainstream solution is to build a variety of shallow structural models, including hidden Markov models [15], Boosting [17] and support vector machines (SVM) [18]. These models are carefully analyzed based on physical and statistical analysis by selecting the threshold parameter. The raw data or processed data are used to divide the gait stage. These methods construct feature engineering to adaptively learn model parameters and obtain hidden relationships and information between historical data. However, gait phase detection is still a challenging problem because the high sampling frequency data collected from sensors always contains the complex nonlinear relationship with multiple components makes it impossible to apply traditional models to analyze sensory data and distinguish walking information in real time.

The emerging deep learning technology has excellent ability to detect complex time relationships. Thanks to breakthroughs in the design and training of model architectures with complex structures composed of multiple processing layers or nonlinear transformations, unprecedented improvements have penetrated many aspects of intelligence, including large-scale visual classification, natural language processing and time series forecasting. Such rapid research progress has also attracted the attention of relevant researchers and companies to build software and hardware to identify the gait stage of snapshots in real life. In particular, convolutional neural networks (CNN) and recurrent neural networks (RNN) have been used to extract the motion characteristics of time-series time data obtained from IMU’s accelerometers and gyroscopes. For example, Omid [19] and others designed a deep convolutional neural network (DCNN) to extract discriminative features from the 2D extended gait cycle and jointly optimize the recognition model in a discriminant manner to complete accurate recognition of the gait phase. Due to their ability to process two-dimensional signals (such as images), most CNNs must convert time-series inertial data into energy images or visually segmented data. This does help to utilize the characteristics of spatial relationships in the gait-phase-recognition, but when processing sequential time series data captured by the IMU sensor, it will obviously ignore the time law and periodic changes, which is often difficult to measure the continuous motion trajectory and extract The quality characteristics of the lower extremities are unrestricted. Therefore, RNN and its improved models including long short-term memory (LSTM) network and gated recurrent unit (GRU) network believe that the current output layer captures a high degree of nonlinearity and time through the time recording sequence and the parameters of the previous hidden layer. Sequential relationship-IMU serial data have attracted wide attention of researchers. Gao et al. [20] used RNN to accurately collect acceleration data to complete the control of the prosthesis. Khokhlova et al. [21] used the LSTM model to classify Normal and Pathologic gait. In addition, CNN can also be combined with traditional machine learning methods, such as SVM [22]. In [22], CNN is used as a feature extractor, and then the extracted features are classified by SVM. At the same time, researchers have found that CNNs tend to ignore temporal continuity when processing only a single time stamp of data [23]. Therefore, it is more common to introduce RNN. The time information of the recording order is passed to the current hidden layer by passing the previous hidden layer state. The LSTM network [24] and GRU network [25] are all RNN. [26] recommended to combine LSTM and CNN for high-dimensional feature representation and character detection. In this way, CNN is usually used as a feature extractor, and then LSTM is used to further process the gesture features extracted by CNN [27]. Gait-phase-recognition was introduced in the comparison of DNN, CNN and RNN in many application scenarios [28].

CNN can use convolutional feature maps to process inertial data, while RNN can also process them as time series. However, research on how to combine these two types of neural networks to achieve better gait-phase-recognition is still lacking. LSTM can mine the timing information in the information better, while CNN can mine the spatial information in the signal better. Therefore, this article combines LSTM and CNN to identify the gait phase. However, if the two are simply combined together, the gradient disappears because the network is too deep, so this paper designed the skip-connection structure and batch normalization layer to alleviate this phenomenon.

Existing research and results show that the FMS-Net model proposed in this paper is effective in gait-phase-recognition. However, most of the data for these works were collected under road conditions under specific walking conditions. Identifying gait phases under complex conditions is still challenging and requires further research.

The rest of the study is organized as follows: The second section introduces data sources and preprocessing techniques and then introduces the gait-phase-recognition model constructed in this paper in detail. Third, evaluate the experimental results of the model through related comparison methods and conduct related discussions. Finally, Section 4 presents our conclusions.

## 2. Materials and Methods

### 2.1. Data Collection

In terms of experimental data, 16 volunteers with body weight ranging from 46 kg to 70 kg and height ranging from 158 cm to 177 cm were selected to collect IMU data. The height and weight distribution of the subjects is shown in Figure 1. The subject’s legs or feet did not have any diseases that could affect normal walking.

With the advancement of sensor processing technology and algorithms, this study selected one IMU modules to collect the corresponding inertial information. Input data in this work only include Lower leg calf acceleration signals. The hardware characteristics required for signal acquisition will be introduced next. To collect lower limb calf acceleration signals, the JY901 nine-axis angle sensor (the type is Uxin Electronics Co., Ltd., Gansu, China) with Kalman filtering algorithm is used in this paper, as shown in Figure 2. There are two communication modes that can be selected: serial port communication and I2C communication. In order to cooperate with the microcontroller, serial port communication is selected for this topic. The TX, RX, VCC and GND pins corresponding to the serial communication are used to connect to the microcontroller. The microcontroller selected is STM32C8T6, which is a 32-bit microcontroller based on the ARM Cortex-M core STM32 series, the program memory capacity is 64 KB, the required voltage is 2 V–3.6 V, and the operating temperature is −40 °C–85 °C; the operating frequency is 72 MHz.

The inertial sensor module is placed outside the lower leg. The arrangement of acceleration sensors for calf monitoring lower limb movement is shown in Figure 3. The system flow of the entire experimental process contains the data collection, processing and application parts. The acceleration resolution of the nine-axis inertial sensor module (MPU9250) used in the experiment is 0.0005 g, the stability of the attitude measurement is 0.05°, and the transmission baud rate is set to 115,200 bps.

During the experiment, all participants were required to walk normally on the same treadmill at a speed of 0.78 m/s, 1.0 m/s and 1.25 m/s for at least 120 s. All participants were asked to normally walk 3 times at each speed. All participants have the same sports environment in the same state. In order to prevent participants from affecting the later movement gait due to continuous exercise, the experiment requires all participants to rest for 2 min after completing the designated walking test each time to alleviate the possible impact of exercise fatigue on walking gait. In addition, when collecting data, it should be noted that we only start saving data after the running speed of the treadmill reaches the set speed. When the treadmill starts to slow down, we stop collecting data and complete the data collection process.

### 2.2. Data Preprocessing

Each input vector contains three acceleration data along the x, y and z directions. Let two sequences of gait data $\vec{a}$ be the input of the network, which is expressed as:(1)$$a_{x}=(a_{x,1},a_{x,2},\ldots ,a_{x,T})$$
(2)$$a_{y}=(a_{y,1},a_{y,2},\ldots ,a_{y,T})$$
(3)$$a_{x}=(a_{z,1},a_{z,2},\ldots ,a_{z,T})$$
(4)$$\vec{a}=(a_{x},a_{y},a_{z})\;$$
where $a_{x},a_{y}$ and $a_{z}$ respectively represent the inertial acceleration in the X, Y and Z directions, and *T* is the length of the input sequence.

According to the above experimental settings, we can obtain the acceleration data curves in the X, Y and Z directions collected by the inertial sensor, as shown in Figure 4.

The human walking process is a cyclical movement, the complete gait cycle is from one-sided heel landing-to-landing again [29]. Although, two phase model recognition systems are sufficient to control active knee orthosis [30]. However, the most widespread method currently relies on four-phase identification technology [31], which are represented as Flat Foot (FF), Heel-Off (HO), Heel Strike (HS) and Swing phase (SW). This gait four-phase detection model has been successfully used to drive ankle-foot orthosis robots [32].

According to previous studies and the scientific nature of the gait phase division, this article also divides the walking cycle into HS, FF, HO and SW phases. During normal walking, the acceleration signal on the calf has a strong periodicity. Studies have shown that the swing phase segment accounts for about 40% of the entire gait cycle and the standing phase accounts for about 60% of the entire gait cycle. According to the previous analysis [1], the schematic diagram of gait cycle division is shown in Figure 5.

### 2.3. FMS-Net Neural Network

The next step is to design an algorithm model to identify the relevant human gait phase through the input acceleration time series vector signal. Since the input acceleration signal is a time series signal, its current signal will have a strong correlation with the previously generated signal, so we need a network model that can mine the internal time series information. Among them, thanks to the design of the hidden layer, the LSTM network can handle timing signals very well. However, LSTM can only mine the timing information in the signal, and it is easy to ignore the spatial information of the signal. Therefore, this paper proposes the fusion of spatiotemporal neural networks. In order to alleviate the gradient disappearing phenomenon in the transmission process, this paper introduces batch normalization (batch norm) layer and skip connection structure. This design can reduce the use of the dropout layer, which can further improve the performance of the network. The entire structure is showed in Figure 6.

As shown in Figure 6, the FMS-Net model for gait-phase-recognition is composed of CNN, LSTM and multiple fully connected layers. CNN can extract important spatial features in the data and reduce the number of network parameters through parameter sharing. In addition, LSTM, as a feature extractor, can obtain the corresponding timing features well. Then, they are followed by two complete fully connected layers, which are used as classifiers. What we need to emphasize is that the “skip-connection structure” proposed in this paper is that the input vector of the first fully connected layer is the superposition of the output vector of the LSTM and the input vector of the entire network, not the addition. The expression is shown in Equation (5).

In addition, in order to prevent the problem of gradient disappearance, batch normalization is used in this paper. Its expression is shown in Figure 6. Learning a deep network is a complicated process. As long as the input layer of the network changes slightly, the network parameters of the subsequent layers will be accumulated and amplified. Once the distribution of input data in a layer of the deep network changes, then this layer of network needs to adapt to learning this new data distribution. During the training process, in order to improve the situation where the data distribution of the middle layer of the network changes, Ioffe et al. [33] introduced batch normalization. The process of batch normalization is shown in Equations (6)–(9). In order for our network to learn to recover the feature distribution that the original network would learn, this learnable reconstruction parameter γ,β is introduced.
(5)$$x_{FC1}=concat\;(o_{lstm},x)=\left[\begin{array}{l} o_{lstm}\\ \;x \end{array}\right]$$
(6)$$\mu_{BN}\leftarrow \frac{1}{m}\sum_{i=1}^{m}x_{i}$$
(7)$$\sigma_{BN}\leftarrow \frac{1}{m}\sum_{i=1}^{m}(x_{i}-\mu_{BN}{)}^{2}$$
(8)$$\overset{\wedge }{x_{i}}\leftarrow \frac{x_{i}-\mu_{BN}}{\sqrt{\sigma_{BN}{}^{2}+\varepsilon }}$$
(9)$$BN_{\gamma ,\beta }(x_{i})\leftarrow \gamma \overset{\wedge }{x_{i}}+\beta$$
where $x_{FC1}{,o}_{lstm},\;x,\;m,\;\mu_{BN}$ and $\sigma_{BN}$ represent the input vector of the first Fully connected layer in the FMS-Net network, the output vector of the LSTM network, the input vector of the FMS-Net network, batch size, batch mean and batch variance, respectively and γ,β are learnable reconstruction parameters.

In this paper, LSTM and CNN is selected as the combined classifier, and some of the network parameters are shown in Table 1. In addition, num_units in the LSTM network is 36, forget_bias is 0.7 and Activation is Relu. The focus of research is the design of the neural network structure. CNN can share weight information through convolution kernel and reduce network parameters. The latter’s LSTM network structure can enhance the learning effect of the network. Finally, specify the final learning rate and set it to 0.05. Finally, the neural network outputs the classification results through the Softmax regression layer.

The FMS-Net algorithm is a model for multiple classification tasks. However, the output of the neural network does not conform to the probability distribution, so it is necessary to convert the output of the neural network into a probability distribution through the Softmax function. The expression of the Softmax function is shown in Equation (10). Then calculate the classification result by Equation (11). Finally, the cross-entropy loss of the model is calculated by Equation (12) and the model parameters are optimized by the gradient descent method.
(10)$$\mathrm{soft}\max(q_{i})=\frac{e^{q\prime_{i}}}{\sum_{i=1}^{n}e^{q\prime_{i}}}$$
(11)O=max(q)
(12)$$l=-\sum_{i=1}^{4}y_{i}\log(q_{i})$$
where, $y_{i}$ denotes the indicative variable (0 or 1), if the category is the same as the sample category, it is 1, otherwise it is 0; $q_{i}$ denotes the predicted probability that the observation sample belongs to category *i*.

For each input vector *x*, the predicted output of the network is $q=(q_{0},q_{1},q_{2},q_{3})$. After Equation (10), the value of $q_{i}$ is between 0 and 1, and the larger the value, the greater the probability that *x* belongs to the real label. Based on the output $q_{i}$, we can get the class label as O.

As can be seen from Equation (12) that, the cross entropy is a positive number. When the probability value of the true label $q_{i}$ in the vector q is smaller, larger difference between $q_{i}$ and $y_{i}$ will result in a larger cross-entropy value. This property will help the convergence of the network in the training.

In order to avoid overfitting, we chose 70% of the sample set for training and 30% of the samples for testing. After using the same training set to train different models 10,000 times, use the same test set to test the trained model, and record the classification accuracy, macro-F value and macro- of each classifier after testing the classification model with the test machine accuracy (AUC). Then evaluate the performance of all models based on these three indicators.

## 3. Results and Discussion

### 3.1. Evaluation Methods

In order to prove the classification performance of the proposed FMS-Net network, we need to draw the corresponding conclusion through corresponding indicators. As we all know, Accuracy is a good comprehensive indicator, which is widely used in evaluation indicators. However, in the classification, it is difficult to characterize the performance of a certain model simply by relying on Accuracy and we must choose other indicators to comprehensively characterize the classification performance of a certain model. In classification problems, commonly used classification performance indicators also include precision, recall and F1. Among them, precision and recall are widely used in the field of information retrieval and statistical classification. These two indicators are used to evaluate the quality of the model results. precision is used to measure the accuracy of the retrieval system. recall is used to measure the recall of the retrieval system. Of course, we hope precision and recall results are as high as possible. Generally speaking, if both precision and recall are high, we can conclude that this model performs well in this classification task. We hope that there is an indicator that can represent the performance of the model in both precision and recall. F1 comprehensively considers the influence of P and R and can comprehensively measure P and R. However, this study studies a multiclassification task and cannot directly use F1. The most direct method is to calculate macro-F1 [34]. Accuracy reflects the ratio of correctly classified samples to total samples. The above evaluation factors are shown in Equations (13)–(18), where *TP*, *TN*, *FP* and *FN* represent true positive, true negative, false positive and false negative, respectively.

In multi-classification tasks, we also have an indicator that is often used to measure classification performance. The more famous is the area under the ROC. The ROC chart was first publicly proposed by Spackman (1989) when performing machine learning and he proved the important role of ROC curve in model evaluation. [35]. In recent years, it was widely used in the fields of machine learning and deep learning. People also realize that simple classification accuracy cannot measure the comprehensive performance and performance of the designed model [29]. We can make the conclusions more reliable by comparing AUC.
(13)$$Accuracy=\frac{TP+TN}{TP+FP+TN+FN}$$
(14)$$P_{i}=\frac{TP}{TP+FP}$$
(15)$$R_{i}=\frac{TP}{TP+FN}$$
(16)$$\;macro-P=\frac{1}{n}\sum_{i=1}^{n}P_{i}$$
(17)$$macro-R=\frac{1}{n}\sum_{i=1}^{n}R_{i}$$
(18)$$macro-F1=\frac{2\times macro-P\times macro-R}{macro-P+macro-R}$$

### 3.2. Results

As shown in Figure 7, Figure 8 and Figure 9, the confusion matrix provides the performance of visual gait sub-phase recognition. The vertical axis of the matrix represents the actual classification category of the test and the horizontal axis represents the corresponding predicted classification category. In addition, in the confusion matrix diagram, “0.0” represents the “HS” phase, “1.0” represents the “FF” phase, “2.0” represents the “HO” phase and “3.0” represents the “SW” phase. These nine matrices are the average recognition results of all subjects under different walking pace conditions. The value in the main diagonal is the proportion of samples correctly classified. As shown in Figure 7, all the confusion matrices, except for the HS phase, perform quite well. The HS phase is mostly incorrectly classified as the FF and SW phases. In order to verify the effectiveness of the proposed recognition model, we implemented two other gait-phase-recognition methods, namely LSTM and LSTM + CNN. The corresponding confusion matrix is shown in Figure 8 and Figure 9. It can be drawn from Figure 8 and Figure 9 that the LSTM and LSTM + CNN models cannot identify the HS phase and directly categorize most of the HS phases into adjacent FF phases by mistake. The LSTM and LSTM + CNN models have achieved good recognition effects on other phases.

From the confusion matrix, we can get Table 2. As shown in Table 2, the macro-F1 of the four groups (HS, FF, HO, SW) of the FSM-Net model differ greatly. When the pace is 0.78 m/s, F1 is, respectively It is 63.7%, 96.4%, 97.6% and 98.8%; when the pace is 1.0 m/s, macro-F1 is 76.6%, 97.6%, 98.2% and 99.0%; when the pace is 1.25 m/s, macro-F1 is It is 54.9%, 97.1%, 97.4% and 98.1%. It can be seen from the above data that FF, HO and SW perform best and obtain a better recognition effect, exceeding 96%). The performance of HS phase recognition is the worst. As for the recognition accuracy of each sub-stage, the SW phase performed best, with the maximum value being the group (99.0%) at a pace of 1.0 m/s. The recognition effect of FF phase and HO phase is also quite good. Obviously, the performance of the HS phase recognition effect is the worst, none of the macro-F1 values reaches 80% and the macro-F1 with the lowest HS phase recognition is only 54.9%. For the LSTM and LSTM + CNN models, macro-F1 for HS phase is 0, showing the worst recognition effect for HS phase; for SW phase recognition, the minimum value of macro-F1 is 97% and 98%.

We also evaluated the performance of the proposed algorithm model and the other two algorithm models in terms of ROC curve. The results are shown in Figure 10, Figure 11 and Figure 12. Through the ROC curve, we can calculate the corresponding macro-AUC and the additional accuracy and macro-F1 statistics to Table 3. It should be noted that “NO-skip” in Table 3 means that the “skip connection” structure is not added to the FMS-Net network. By comparing the FMS-Net network before and after adding the “skip connection” structure, it can be concluded that the skip connection structure has a certain improvement effect on the recognition performance of the FMS-Net network. It is also worth noting that our method obtained a higher macro-F1 value and accuracy, higher than all other groups method.

### 3.3. Discussion

#### 3.3.1. Acceleration System Analysis

Gait analysis provides an opportunity to assess walking behavior. Gait analysis can be used for various applications, such as rehabilitation, clinical diagnosis and physical activity [36]. The acceleration signal generated when the human calf is walking on a flat ground is a regular signal, and this information can be extracted by using the IMU. Although the FMS-Net has shown certain effectiveness in the detection of gait events, it still needs further optimization in the future. In this study, the IMU needs to be placed on the designated position of each subject’s calf. However, due to each subject’s height, weight, gender, walking habits, etc., the sensor cannot be accurately placed at the designated location, and can only be installed at an approximate designated location, which requires further study.

The IMU used in this article is the JY901 sensor. The JY901 sensor uses Kalman filtering to filter the collected data better, filter out redundant noise and ensure the quality of the transmitted signal. In addition, the data output rate of the JY901 sensor is 200 HZ, which can ensure the real time nature of the later signal transmission and avoid the occurrence of repetitive signals at the signal receiving end. The measurement range of acceleration is between −16 g and 16 g, and this range can fully meet the needs of this study. It can be seen by observing Figure 4 that the acceleration data collected by the experiment is between −1 g and 2 g, which is completely within the measurable range of the JY901 sensor. 2.4-G wireless communication adopted between JY901 sensor and host computer. The signal collecting terminal is separated from the signal receiving terminal to avoid the signal cable from having an additional effect on the coordination of the body.

In order to increase the variability of the experiment, this study asked each subject to perform three different pace experiments on a treadmill and let them walk three times at each different pace. Although the collected data have certain differences, there is still a big gap from the complex walking state in reality, and further research is needed.

In reviewing the literature, regarding gait events, the ANN algorithm model was used to achieve an 82.2% recognition accuracy rate for the IMU under different walking conditions [37]. When using the IMU to classify the five gait phases, an accuracy of 82% can be achieved [34]. In addition, gait-phase-recognition is very important for the development of calf assist devices because they are strongly related to gait events [35]. In order to propose an acceleration data acquisition system that can be applied to the masses, we tested the recognition effect of the FMS-Net algorithm proposed in this study on unlearned acceleration signal data. This study found that the proposed FMS-Net can successfully predict gait events for test set, and the accuracy of phase recognition for HS, FF, HO and SW is up to 96%, and based on acceleration signals, detection of HS, FF, HO and SW phase seems to be reliable.

#### 3.3.2. Gait Phase Detection

The core technology of gait-phase-recognition system is the design of recognition algorithm model. This paper proposes the FMS-Net and uses it to detect the HS, FF, HO and SW phases. According to the results obtained in this study, the acceleration signal has relatively high stability when walking on the ground, which satisfies the research in this study. In this study, the FMS-Net algorithm combines LSTM, CNN, skip-connection and other structures. The algorithm model has certain complexity. Although good results were achieved, it should continue to be optimized in the direction of lightweight networks in the future.

When walking on flat ground, the acceleration signal on the human calf is a typical time-series signal. LSTM is a classic algorithm model for processing time-series signals and CNN is a typical algorithm for extracting spatial information from signals. LSTM + CNN directly combines the two, but it is easy to cause the gradient to disappear as the depth becomes larger. This study takes full advantage of the advantages of LSTM in processing timing signals and CNN’s convolution operation in extracting spatial features and uses skip-connection structure and batch normalization to solve the problem of deep gradient disappearance and design FMS-Net algorithm model. From the results of Figure 7, Figure 8 and Figure 9, the three models show good recognition effect on the FF, HO and SW phases, but LSTM and LSTM + CNN cannot accurately identify HS. Although LSTM + CNN is still unable to identify the HS phase, it is superior to LSTM in the recognition of the other three phases compared to the LSTM algorithm. The FMS-Net algorithm by adding skip-connection structure has been further improved compared with LSTM + CNN. Through Figure 9, we can see that the FMS-Net algorithm has further improved the phase recognition of HS and can recognize most of the HS phase. Although the FMS-Net algorithm can identify part of the HS phase, it is still lower than 80%, so further optimization and improvement are still needed. By observing Figure 10, Figure 11 and Figure 12, it can be seen that the AUC of the HS phase of the FMS-Net algorithm is also superior to LSTM and LSTM + CNN. It can be seen from Table 1 that the LSTM and LSTM + CNN recognition of HS phase F1 is 0, while the average F1 of FMS-Net algorithm for HS phase recognition at three paces is 65.1%, which may be better than HS During the transmission of phase data, the gradient disappears. It can be seen from Table 2 that the performance of the three is not much different in accuracy, FMS-Net is the best, LSTM + CNN is the second and CNN is the worst. However, the performance of macro-F1 is quite different. The average macro-F1 of FMS-Net at three paces is 89.5%, while the average macro-F1 of LSTM + CNN and CNN at three paces is 72.4% and 71.4%. In terms of AUC, FMS-Net also performs best. As can be seen from Table 3, the recognition performance of FMS-Net when walking at a speed of 1.0 m/s is the best among the three. The reason for the result may be that 1.0 m/s is relatively close to the normal walking speed, but this conclusion still needs to be proved by adding more control groups. In the future, we need to set more walking speed control experiments to obtain more reliable conclusions.

Even if FMS-Net shows its usefulness in classifying acceleration signals detected by gait events, other deep-learning methods need to be used for further evaluation. Future work should improve classification accuracy by improving feature extraction and gait-phase-recognition algorithms. In this study, it is considered acceptable to install a wearable inertial sensor module only on the lower leg compared to other wearable sensors. However, in fact, when the subject wears the sensor for a long time, it may have a potential impact on the subject’s gait, which requires further exploration. In the future, we will try to use new gait functions (for example, gait dynamic images [38,39]) instead of neural network input original x, y, z to verify its effect on gait-phase-recognition, which may be one of the future jobs.

## 4. Conclusions

In order to meet the application of gait-phase-recognition technology in the control of lower extremity dynamic exoskeleton, this study propose a low-cost, easy to implement and efficient IMU-based gait sub-phase recognition system. First, we constructed a wireless calf acceleration signal acquisition device. Then, we preprocess the collected data in order to train the classifier for the subsequent use of the data set. Finally, a novel classifier FMS-Net applying seamlessly combining LSTM and CNN models by applying the skip-connection structure is established to extract acceleration signal features and predict gait sub-phase. Experiments and discussions prove that the FMS-Net method has better classification accuracy with the macro-F1 up to 96%, which is superior to other integrated algorithm models. The results show that the proposed method can effectively perform gait-phase-recognition, which lays a solid foundation for the application of gait-phase-recognition technology in the control of lower extremity dynamic exoskeleton.

## Figures and Tables

**Figure 1 ijerph-17-05633-f001:**
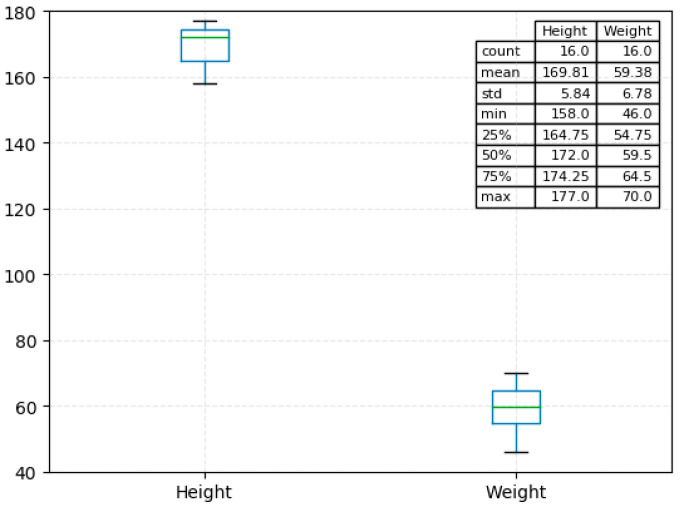
Information about volunteers participating in this experiment.

**Figure 2 ijerph-17-05633-f002:**
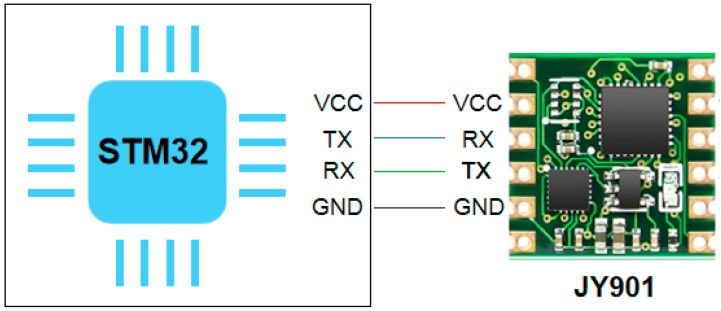
Schematic diagram of inertial sensors (IMU) connected to single chip microcomputer.

**Figure 3 ijerph-17-05633-f003:**
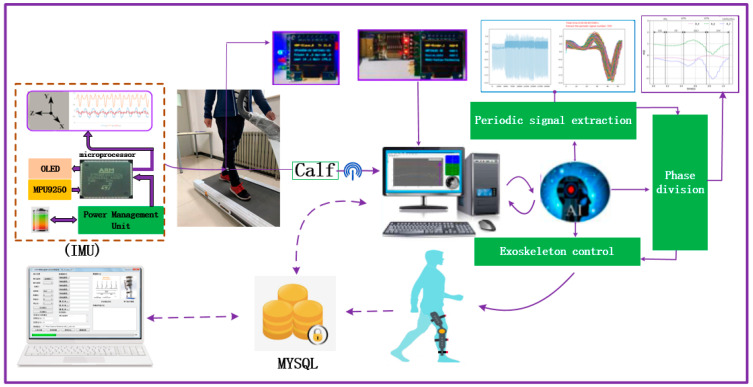
Human gait-information acquisition system.

**Figure 4 ijerph-17-05633-f004:**
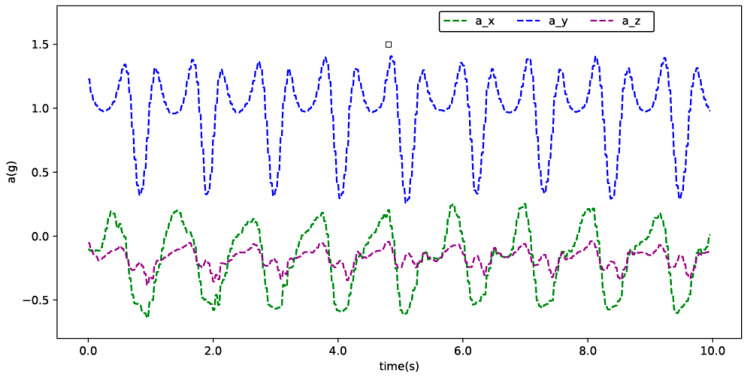
Acceleration data collected under the calf. a_x, a_y and a_z represent the acceleration data in the *X*-axis, *Y*-axis and *Z*-axis directions collected by the experimental equipment, respectively.

**Figure 5 ijerph-17-05633-f005:**
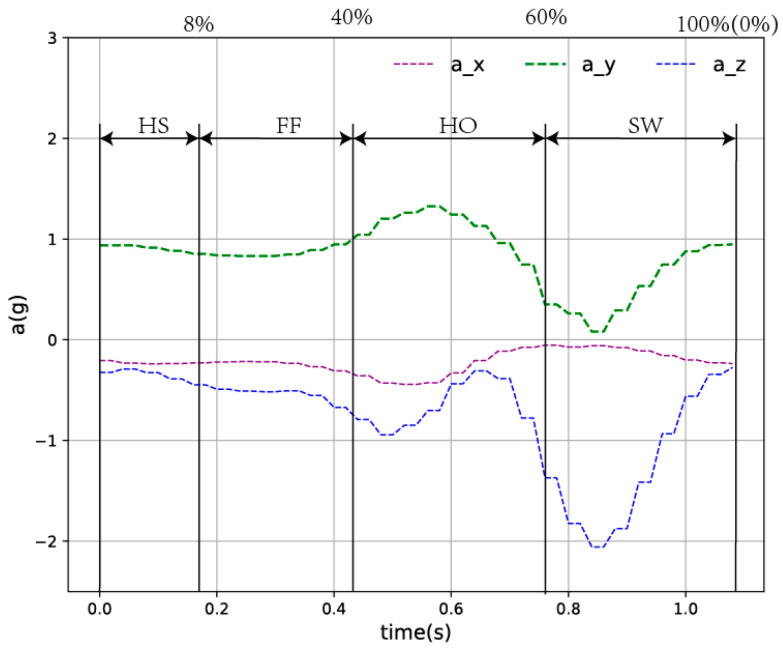
Phase-division diagram of gait.

**Figure 6 ijerph-17-05633-f006:**
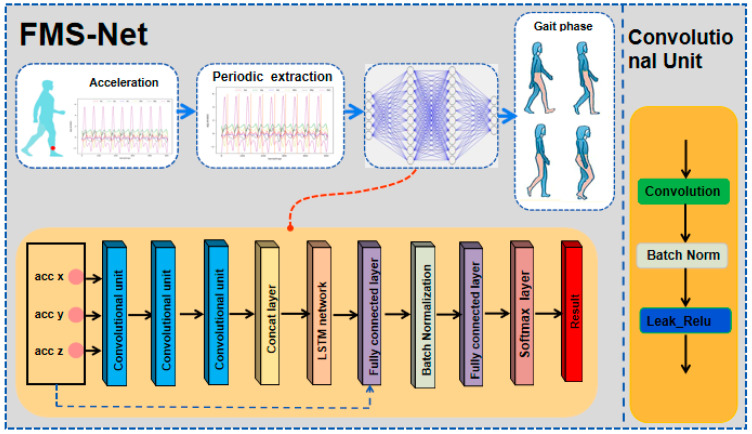
Proposed network (FMS-Net) architecture.

**Figure 7 ijerph-17-05633-f007:**
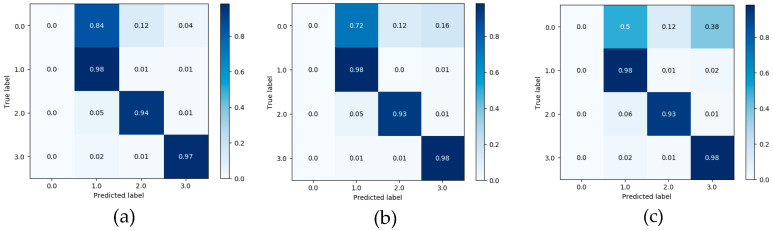
Confusion matrix obtained from LSTM classifier at three different paces: 0.78 m/s (**a**), 1.0 m/s (**b**) and 1.25 m/s class (**c**).

**Figure 8 ijerph-17-05633-f008:**
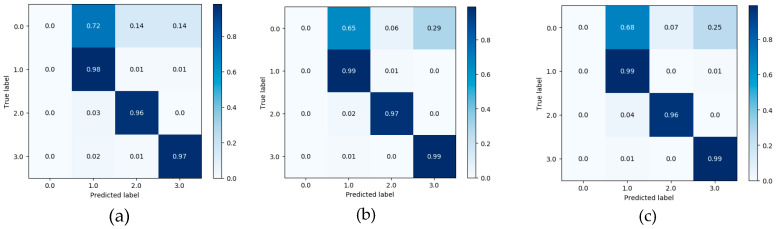
Confusion matrix obtained from LSTM+CNN classifier at three different paces: 0.78 m/s (**a**), 1.0 m/s (**b**) and 1.25 m/s class (**c**).

**Figure 9 ijerph-17-05633-f009:**
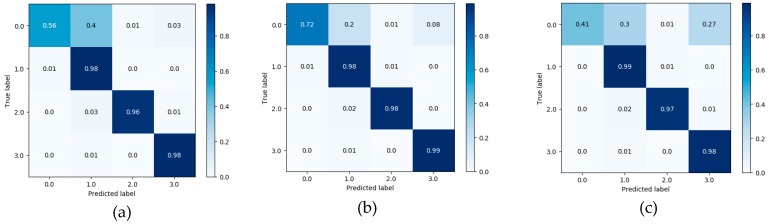
Confusion matrix obtained from FMS-Net classifier at three different paces: 0.78 m/s (**a**), 1.0 m/s (**b**) and 1.25 m/s class (**c**).

**Figure 10 ijerph-17-05633-f010:**
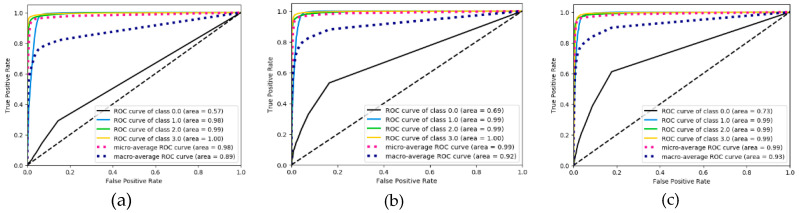
ROC curve performance derived from LSTM classification under three pace settings: 0.78 m/s (**a**), 1.0 m/s (**b**) and 1.25 m/s classes (**c**).

**Figure 11 ijerph-17-05633-f011:**
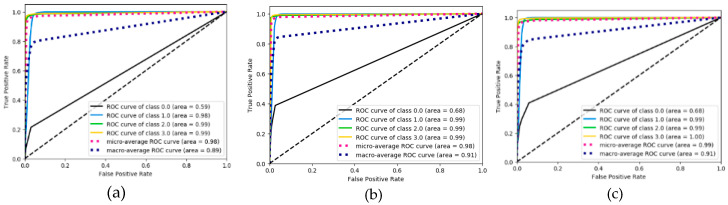
ROC curve performance derived from LSTM+CNN classification under three pace settings: 0.78 m/s (**a**), 1.0 m/s (**b**) and 1.25 m/s classes (**c**).

**Figure 12 ijerph-17-05633-f012:**
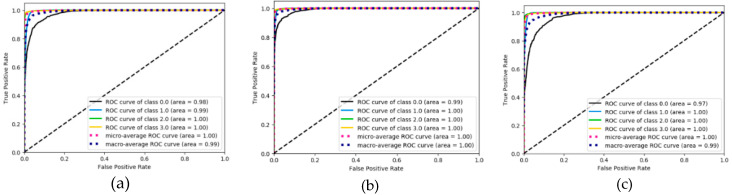
ROC curve performance derived from FMS-Net classification under three pace settings: 0.78 m/s (**a**), 1.0 m/s (**b**) and 1.25 m/s classes (**c**).

**Table 1 ijerph-17-05633-t001:** Parameter setting of FMS-Net structure.

CNN					FC_Layer	
Kernel_Size	Activation	Filters	Stride	Feature_Map	Num_Units	Activation
4 × 4	relu	20	1	2 × 2 × 20	160	leaky_relu
1 × 1	relu	5	1	1 × 1 × 5	60	leaky_relu
1 × 1	relu	1	1	1 × 1 × 1	4	leaky_relu

**Table 2 ijerph-17-05633-t002:** Summary of classification performance of different models at unsynchronized speed.

Model	Speed	0.78 m/s				1.0 m/s				1.25 m/s			
	Phase	HS	FF	HO	SW	HS	FF	HO	SW	HS	FF	HO	SW
LSTM	Precision (%)	0	89.0	94.6	98.3	0	90.5	95.6	97.8	0	91.8	95.2	96.4
	Recall (%)	0	97.9	93.9	97.2	0	98.5	93.4	97.8	0	97.6	92.9	98.0
	F1 (%)	0	93.3	94.3	97.8	0	94.3	94.5	97.8	0	94.6	94.0	97.2
LSTM+CNN	Precision (%)	0	91.3	93.8	98.5	0	92.8	97.6	97.9	0	91.9	97.2	98.1
	Recall (%)	0	98.2	96.3	97.4	0	99.0	97.3	98.9	0	99.2	95.7	98.7
	F1(%)	0	94.6	95.1	98.0	0	95.8	97.4	98.4	0	95.4	96.4	98.4
	Precision (%)	73.8	94.5	98.2	99.2	82.3	96.8	98.5	99.1	82.9	95.6	97.4	97.8
FMS-Net	Recall (%)	56.0	98.3	97.0	98.4	71.6	98.4	97.9	98.9	41.0	98.5	97.4	98.3
	F1 (%)	63.7	96.4	97.6	98.8	76.6	97.6	98.2	99.0	54.9	97.1	97.4	98.1

**Table 3 ijerph-17-05633-t003:** Summary of classification performance for different training functions.

Pace	Training Function	Classification Rate		
		Accuracy (%)	Macro-F1 (%)	Macro-AUC
0.78 m/s	LSTM	94.2	71.3	0.89
	LSTM+CNN	95.1	71.9	0.89
	FMS-Net	96.7	88.9	0.99
	NO-skip	96.5	88.0	0.99
1.0 m/s	LSTM	94.7	71.6	0.92
	LSTM+CNN	96.0	72.9	0.91
	FMS-Net	97.8	92.8	1.0
	NO-skip	97.2	91.7	0.99
1.25 m/s	LSTM	94.4	71.4	0.93
	LSTM+CNN	95.7	72.5	0.91
	FMS-Net	96.8	86.9	0.99
	NO-skip	96.6	85.9	0.99

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
