# Peer review of "An Acceleration Based Fusion of Multiple Spatiotemporal Networks for Gait Phase Detection"

_ijerph, 2020, doi:10.3390/ijerph17165633_

Round 1

Reviewer 1 Report

Several language mistakes are present in this acceptable work (eg feets instead of feet  and so on) there are also aims some editing errors (I suppose ^{....} in formulas embedded into the text). The content is amazing and the idea of fusion is rather new from this particular point of view. Errors in measurement seems to be acceptable é as promising. This reviewer encourages the publication after an extensive correction of the presentation style 

Author Response

Responds to the reviewers’ comments

Dear Editors and Reviewers:

Thank you for your letter and for the reviewers’ comments concerning our manuscript entitled "An acceleration based Fusion of Multiple Spatiotemporal Networks for gait phase detection" (Manuscript ID: 831003 (Research Article)). Those comments are all valuable and very helpful for revising and improving our paper, as well as the important guiding significance to this research. We have studied comments carefully and have made correction which we hope meet with approval. We tried our best to improve the manuscript and made some changes in the manuscript according to your suggestions. These changes are marked in highlighted style, which will not influence the content and framework of the manuscript.

We appreciate for editors and reviewers’ warm work earnestly, and hope that the correction will meet with approval. The main corrections in the paper and the responds to the reviewers’ comments are as following:

Reviewer #1:

General comments:

Several language mistakes are present in this acceptable work (eg feets instead of feet  and so on) there are also aims some editing errors (I suppose ^{....} in formulas embedded into the text). The content is amazing and the idea of fusion is rather new from this particular point of view. Errors in measurement seems to be acceptable é as promising. This reviewer encourages the publication after an extensive correction of the presentation style 

Response: 

First of all, I am very grateful to the reviewers for their affirmation of our paper and their valuable comments and suggestions. We carefully studied the comments and suggestions of the reviewers, and made many changes very carefully according to the reviewers' requirements.

  1. According to the reviewer's request, we asked English speakers to retouch the paper. We hope this description will be more accurate and familiar to you and readers.
  2. According to the reviewers’comments, we have revised the format of all formulas in the paper, hoping to satisfy the reviewers.

Thank you very much. 

Best wishes,

Tao Zhen

Reviewer 2 Report

The conclusion of "Experimental results show that this method has better classification accuracy than other methods..." is not supported by the evidence in the paper.  The LSTM and LSTM + CNN models proposed were incapable of identifying Heel Strike.   The authors further write "FMS-Net algorithm can identify part of the HS phase, it is still lower than 80%, so further 439 optimization and improvement are still needed"...which does not support the conclusion in the abstract that the method has better classification accuracy than other methods. 

- There is no mention of review nor approval of the study by an Institutional Research Ethics Board. 

-Further cross validation should be implemented - e.g. 5 fold or 10 fold cross-validation or leave-one-out type of approach.

- 16 subjects is really too low a number to have much confidence that the presented results can generalize. Different stratifications of the data are not mentioned nor tested.  How large is the dataset? The reader can only infer that the 16 subjects each walked one 2 minute trial at each of the three proposed speeds? 

- The authors switch between using the first person plural and first person singular, which is awkward to read. 

- The manuscript contains many similarities with ref [1] and although the results from [1] are superficially compared in this manuscript, why was the approach proposed in [1] not attempted with this dataset (different sensors/sensor location). 

-At the moment, the manuscript can be summarized as "we combined a bunch of black box NN approaches and implemented the proposed combination on a limited dataset and got mediocre results".  The paper does not inform on (i) optimal sensor placement (ii) optimal sensor, optimal length of walking trial (iii) robust algorithms (iv) thoughtful validation of the proposed approach.   

Hence, it is difficult to decipher what the real contribution of the paper is. 

Author Response

Responds to the reviewers’ comments

Dear Editors and Reviewers:

Thank you for your letter and for the reviewers’ comments concerning our manuscript entitled "An acceleration based Fusion of Multiple Spatiotemporal Networks for gait phase detection" (Manuscript ID: 831003 (Research Article)). Those comments are all valuable and very helpful for revising and improving our paper, as well as the important guiding significance to this research. We have studied comments carefully and have made correction which we hope meet with approval. We tried our best to improve the manuscript and made some changes in the manuscript according to your suggestions. These changes are marked in highlighted style, which will not influence the content and framework of the manuscript.

We appreciate for editors and reviewers’ warm work earnestly, and hope that the correction will meet with approval. The main corrections in the paper and the responds to the reviewers’ comments are as following:

Reviewer #3:

General comments:

  • The conclusion of "Experimental results show that this method has better classification accuracy than other methods..." is not supported by the evidence in the paper.  The LSTM and LSTM + CNN models proposed were incapable of identifying Heel Strike.   The authors further write "FMS-Net algorithm can identify part of the HS phase, it is still lower than 80%, so further 439 optimization and improvement are still needed"...which does not support the conclusion in the abstract that the method has better classification accuracy than other methods. 

Response: Compared with other algorithm models, the FMS-Net algorithm has an excellent performance in HS phase recognition, resulting in a higher F1 of the FMS-Net algorithm. And through the accuracy index in the result, it is not difficult to see that the model proposed in this article is indeed better than other models in recognition effect.

  • - There is no mention of review nor approval of the study by an Institutional Research Ethics Board. 

Response: The research complies with Institutional Research Ethics Board regulations. I have explained this in the "Conflicts of Interest" section in the original text.

  • -Further cross validation should be implemented - e.g. 5 fold or 10 fold cross-validation or leave-one-out type of approach.

Response: The three-fold cross-validation method actually used in this experiment. Sorry for not explaining this. Due to resource constraints during the epidemic, no more tests were carried out, but I believe that the experimental results have a certain degree of credibility. If the reviewer feels that the experiment still needs to be done, we will try to find a way to complete the corresponding experiment as soon as possible.

  • 16 subjects is really too low a number to have much confidence that the presented results can generalize. Different stratifications of the data are not mentioned nor tested.  How large is the dataset? The reader can only infer that the 16 subjects each walked one 2 minute trial at each of the three proposed speeds? 

Response: I need to explain a little bit about the data set. In fact, the data set is not very small. Each subject completed the relevant experiments at three different paces. At each same pace, subjects need to perform the same three experiments, so each subject needs to perform nine walk experiments. Each experiment needs to collect about 120 seconds of data, and the collection frequency is 50HZ, so a total of about 860,000 data are collected. When dealing with data, we actually spent a lot of energy.

  • The authors switch between using the first person plural and first person singular, which is awkward to read

Response: Thank you very much to the reviewer for raising this question. Based on the reviewers’ comments, I have changed the first-person singular to the first-person plural.

  • The manuscript contains many similarities with ref [1] and although the results from [1] are superficially compared in this manuscript, why was the approach proposed in [1] not attempted with this dataset (different sensors/sensor location). 

Response: Thank you very much to the reviewer for raising this question. [1] uses a voting weighted ensemble algorithm. She uses three parallel networks, which takes a lot of time. This paper designs a space-time fusion algorithm, which greatly reduces the amount of calculation. We hope that the designed model can develop in the direction of light weight and effectiveness. If the reviewer feels it is necessary to add a controlled trial, we will arrange the trial and complete it as soon as possible.

  • At the moment, the manuscript can be summarized as "we combined a bunch of black box NN approaches and implemented the proposed combination on a limited dataset and got mediocre results".  The paper does not inform on (i) optimal sensor placement (ii) optimal sensor, optimal length of walking trial (iii) robust algorithms (iv) thoughtful validation of the proposed approach.   

Response: Previously, most scholars used multiple IMUs for gait phase recognition, but too many sensors would cause a lot of inconvenience to the wearer. Moreover, the sensor may move during the transmission of the human body. Therefore, this article tried to install sensors only on the calf for phase recognition, and achieved good results. Of course, other parts of the body, such as hip joints, chest, feet and other parts of the test still need to be further studied. In the future, we will conduct research on other parts. The sensor collects 120S data during each test. However, it needs to collect 3 times at the same pace, which is equivalent to collecting 360S data at the same pace. The test results prove that data for such a long time It has been able to meet the needs of the test. We will further optimize the experimental process and algorithm model. Thank you very much for your valuable comments.

  • Hence, it is difficult to decipher what the real contribution of the paper is. 

Response: The current gait phase recognition mostly uses multiple IMUs for recognition, which will cause a lot of inconvenience to the wearer and information processing. This article tried to use an IMU for phase recognition, and achieved good results. Previous deep learning methods extracted spatial and temporal features in isolation, while ignoring the inherent correlation in high-dimensional space, which limited the accuracy of a single model. This paper proposes an algorithm model based on the fusion of multiple temporal and spatial networks. Experimental results show that the algorithm model has certain potential in gait phase recognition.

Thank you very much for taking the time to review this paper during your busy schedule. I wish you good health, successful scientific research and a happy life.

Reviewer 3 Report

The authors propose a novel gait phase detection method called FMS-Net.

This paper has a sufficient description of the proposed method, and the method seems to be effective for the future application of the exoskeleton robot control.

The reviewer considers the submitted paper will have the potential to be accepted to the journal.

However, some minor revisions will be needed for publication.

Please check the following comments.

Comments

1.

In the last paragraph of Section 1, the authors’ description seems to have two Section 4 (“Fourth, …” and “Finally, Section 4 …” ), but actually, the paper has only four sections (not five). Please revise the description.

2.

Please review the equation and variables in the whole paper. Some notations of the equations and variables seem to be not good. The same variables should be written in the same font in the equation, figure, table, and text.

For example, line 199, 202, 203, Figure captions, etc.

3.

Figure 5 shows an example of the phase division of one gait cycle data. How do the authors give the correct gait phases? For example, the correct phases are given by the synchronized recorded video?

4.

Please describe the shortened name “FMS-Net” at the first appearance in the abstract and the main text. FMS-Net is “Fusion of Multiple Spatiotemporal Networks”?  

5.

In line 238, “As shown in Figure 5” is “As shown in Figure 6”?

6.

In lines 242 to 244, the sentences “What I want … , not addition” is the description of the skip-connection structure? If so, please emphasize that.

7.

In line 252, “… is shown in Equs. (5) to (9)” is “… is shown in Equs. (6) to (9)”?

8.

In equation (5) to (9), many variables are not defined, for example, x_FC1, o_dnn, o_gru, x_i, \hat{x_i}, \epsilon, and BN_gamma,beta.

9.

In lines 273 to 276 seem to be a misprint since these are the same sentences of the before.

10.

Equations (11) and (12) have no citation in the main body text before that.

One of the solutions is that the sentence in line 272 to 273 “… in equation (10)” is revised as “… in equations (10) to (12)”.

11.

In line 283, “After Equ.(2),” is “After Equ.(10)”?

12.

In line 295, section number seems to be wrong, and Section 3 will be correct.

13.

Equation numbers of (16) to (21) will be wrong, and (13) to (18) will be correct.

14.

In line 343, “The confusion matrix (Figure 7, …” is “The confusion matrix Figure 7, …”?

15.

In Table 2, Model name “FSM-Net” is “FMS-Net”?

16.

In Section 3.3 Discussion, the descriptions “This study proves … of the algorithm” have been already described in the previous sections, and these sentences will not be needed.

17.

In section 3.3.2, there are some typo “GFM-Net”.

18.

In line 454, “1 Future work …” seems strange.

Author Response

Responds to the reviewers’ comments

Dear Editors and Reviewers:

Thank you for your letter and for the reviewers’ comments concerning our manuscript entitled "An acceleration based Fusion of Multiple Spatiotemporal Networks for gait phase detection" (Manuscript ID: 831003 (Research Article)). Those comments are all valuable and very helpful for revising and improving our paper, as well as the important guiding significance to this research. We have studied comments carefully and have made correction which we hope meet with approval. We tried our best to improve the manuscript and made some changes in the manuscript according to your suggestions. These changes are marked in highlighted style, which will not influence the content and framework of the manuscript.

We appreciate for editors and reviewers’ warm work earnestly, and hope that the correction will meet with approval. The main corrections in the paper and the responds to the reviewers’ comments are as following:

Reviewer #3:

General comments:

  • In the last paragraph of Section 1, the authors’ description seems to have two Section 4 (“Fourth, …” and “Finally, Section 4 …” ), but actually, the paper has only four sections (not five). Please revise the description.

Response: I am very grateful to the judges for correcting this. We have already corrected this error in the original text.

  • Please review the equation and variables in the whole paper. Some notations of the equations and variables seem to be not good. The same variables should be written in the same font in the equation, figure, table, and text.

For example, line 199, 202, 203, Figure captions, etc.

Response: Based on the reviewers’ comments, we have edited the formulas and variables in the paper. Please review the paper.

  • Figure 5 shows an example of the phase division of one gait cycle data. How do the authors give the correct gait phases? For example, the correct phases are given by the synchronized recorded video?

Response: At present, the mainstream method of gait phase marking is to record video, and then find the data based on the display of the video. The method is simple and direct. But it is not accurate enough and can only be roughly estimated. There is another way to determine by foot pressure signal. Since our team has been studying to determine the phase through the foot pressure signal before, we have a complete set of foot pressure signal acquisition equipment. Therefore, when I collect IMU signals, I also collect foot pressure signals, so that I can easily find periodic signals and cut them accordingly to complete the labeling.

In addition, we have now found another better method. By establishing the human body motion dynamics model and importing the collected angle data into the model, we can observe the motion acceleration data of any point of the human body. However, in the modeling process, it must be noted that the two coordinate systems must be consistent.

  • Please describe the shortened name “FMS-Net” at the first appearance in the abstract and the main text. FMS-Net is “Fusion of Multiple Spatiotemporal Networks”?  

Response: I am very sorry that my writing mistakes caused the reviewers' doubts in reading. We have made changes in the abstract, FMS-Net is "Fusion of Multiple Spatiotemporal Networks".

  • In line 238, “As shown in Figure 5” is “As shown in Figure 6”?

Response: Due to my mistake, "Figure 6" was written as "Figure 5". I have corrected this error in the original text.

  • In lines 242 to 244, the sentences “What I want … , not addition” is the description of the skip-connection structure? If so, please emphasize that.

Response: Lines 242 to 244 are the description of "skip-connection structure". I have revised the original text based on the reviewer’s suggestions. Thank you very much for your suggestions.

  • In line 252, “… is shown in Equs. (5) to (9)” is “… is shown in Equs. (6) to (9)”?

Response: I'm very sorry, this is my mistake again. "... is shown in Equs. (5) to (9)" should be "... is shown in Equs. (6) to (9)". I have corrected this error.

  • In equation (5) to (9), many variables are not defined, for example, x_FC1, o_dnn, o_gru, x_i, \hat{x_i}, \epsilon, and BN_gamma,beta.

Response: The variable definitions in equation (5) to (9) must be listed after equation (9). In addition, due to my mistakes, there were certain writing errors in the equation , and I have revised them again. Thank you very much for your comments.

  • In lines 273 to 276 seem to be a misprint since these are the same sentences of the before.

Response: Thank you very much to the reviewer for pointing out this obvious error. I have deleted the duplicate part in the original text.

  • Equations (11) and (12) have no citation in the main body text before that.

One of the solutions is that the sentence in line 272 to 273 “… in equation (10)” is revised as “… in equations (10) to (12)”.

Response: I am very sorry, but due to my carelessness, I forgot to describe equations (11) and (12). I have modified the original text here, please review it.

  • In line 283, “After Equ.(2),” is “After Equ.(10)”?

Response:

  • In line 295, section number seems to be wrong, and Section 3 will be correct.

Response: Yes. Thank you very much for your correction.

  • Equation numbers of (16) to (21) will be wrong, and (13) to (18) will be correct.

Response:Thank you very much for your correction. I have seriously modified the description here in the original text.

  • In line 343, “The confusion matrix (Figure 7, …” is “The confusion matrix Figure 7, …”?

Response: Thank you very much for your correction. "The confusion matrix (Figure 7, Figure 8 and Figure 9..." should be "The confusion matrix (Figure 7, Figure 8 and Figure 9)". I have modified the original text.

  • In Table 2, Model name “FSM-Net” is “FMS-Net”?

Response: Yes. This is my mistake again, I have revised the original text.

  • In Section 3.3 Discussion, the descriptions “This study proves … of the algorithm” have been already described in the previous sections, and these sentences will not be needed.

Response: Thank you very much for your suggestion. According to your suggestion, I have deleted this part of the description.

  • In section 3.3.2, there are some typo “GFM-Net”.

Response:Thank you very much for your correction. I have changed "GFM-Net" to "FMS-Net".

  • In line 454, “1 Future work …” seems strange.

Response: This is my mistake again, and I am very sorry. "1 Future work …" should be "Future work …". I have corrected this error. Thank you very much for taking the precious time from your busy schedule to review my paper and for your patience to find out so many detailed errors.

Thank you again. I wish you good health, successful scientific research and a happy life.

Reviewer 4 Report

  1. How are the ground truth label generated?
  2. In FMS-Net, how important is the skip connection? Can you run experiment show the different without skip connection?
  3. Your accuracy rate is 96%, which is two percetp lower than Zhen et al. Are these two approaches evaluated on the same data?  They can not be compared on two different datasets. 
  4. What do you mean "unlearned data"?  Are you referring to testing data?
  5. avoid non-scientific subjective writing, such as "I have alwasys been interested"

Author Response

Dear Editors and Reviewers:

Thank you for your letter and for the reviewers’ comments concerning our manuscript entitled "Walking gait phase detection based on acceleration signals using voting-weighted integrated neural network" (Manuscript ID: 4760297.v1 (Research Article)). Those comments are all valuable and very helpful for revising and improving our paper, as well as the important guiding significance to this research. We have studied comments carefully and have made correction which we hope meet with approval. We tried our best to improve the manuscript and made some changes in the manuscript according to your suggestions. These changes are marked in highlighted style, which will not influence the content and framework of the manuscript.

We appreciate for editors and reviewers’ warm work earnestly, and hope that the correction will meet with approval. The main corrections in the paper and the responds to the reviewers’ comments are as following:

General comments:

  • How are the ground truth label generated?

Response: Now the mainstream tags are produced by recording videos. The method is simple and direct. But it can only be roughly estimated, not accurate enough. There is another way to determine by foot pressure signal. Since our team has been studying to determine the phase through the foot pressure signal before, we have a complete set of foot pressure signal acquisition equipment. Therefore, when I collect IMU signals, I also collect foot pressure signals, so that I can easily find periodic signals and cut them accordingly to complete the labeling.

In addition, we have now found another better method. By establishing the human body motion dynamics model and importing the collected angle data into the model, we can observe the motion acceleration data of any point of the human body. However, in the modeling process, it must be noted that the two coordinate systems must be consistent.

  • In FMS-Net, how important is the skip connection? Can you run experiment show the different without skip connection?

Response: According to the reviewer's opinion, I have conducted related experiments and revised the original text. From the experimental results, the "skip connection" structure can improve the model's recognition effect to a certain extent.

  • Your accuracy rate is 96%, which is two percetp lower than Zhen et al. Are these two approaches evaluated on the same data?  They can not be compared on two different datasets. 

Response: Thank you very much to the reviewer for raising this question. I have realized that there is a problem with my statement, and I have deleted this unscientific statement. Thank you very much for your reminders.

  • What do you mean "unlearned data"?  Are you referring to testing data?

Response: Yes, "unlearned data" refers to the "test set".

  • avoid non-scientific subjective writing, such as "I have alwasys been interested"

Response: Thank you very much for your valuable advice. We re-read the paper according to your request and changed the similar subjective expression.

Round 2

Reviewer 2 Report

Thank you for the revised manuscript.  The paper is little more clear now. Please

  • remove instances of first person (there are still "I xxx" in the paper)
  • please explain in the manuscript how the 'ground truth' for establishing the gait phases were determined. This was explained at length (although somewhat confusingly) in the response to review but I'm not sure if it made itself into the manuscript.
  • In comment about approval from ethics review board, please do not start a sentence with an "And".

EDITOR: please correct/check English language.

Author Response

General comments:

1.remove instances of first person (there are still "I xxx" in the paper) 

Response: 

Thank you very much for your valuable comments. I re-read the paper and found that there is indeed a "...I..". I have modified this, please review it.

2.please explain in the manuscript how the 'ground truth' for establishing the gait phases were determined. This was explained at length (although somewhat confusingly) in the response to review but I'm not sure if it made itself into the manuscript.

Response: Thank you very much for the comments and suggestions of the reviewers. In order to echo the previous work, we added corresponding references in the gait phase division part. Thank you very much for your comments and suggestions.

  1. In comment about approval from ethics review board, please do not start a sentence with an "And".

Thanks to the reviewers for their valuable suggestions. I have deleted "And" according to the reviewers' suggestions.

Thank you very much. 

Best wishes,

Tao Zhen

Reviewer 4 Report

Instead of using raw x,y,z as input to your neural network, did you try gait feature, such as Gait Dynamic Images [1][2], to see if that helps accuracy for your task?  This could be one of the future work.

[1]Zhong, Y., Deng, Y., & Meltzner, G. (2015, September). Pace independent mobile gait biometrics. In 2015 IEEE 7th International Conference on Biometrics Theory, Applications and Systems (BTAS) (pp. 1-8). IEEE.

[2] Zhong, Y., & Deng, Y. (2014, September). Sensor orientation invariant mobile gait biometrics. In IEEE international joint conference on biometrics (pp. 1-8). IEEE.

Author Response

General comments:

  • Instead of using raw x,y,z as input to your neural network, did you try gait feature, such as Gait Dynamic Images [1][2], to see if that helps accuracy for your task?  This could be one of the future work.

 [1]Zhong, Y., Deng, Y., & Meltzner, G. (2015, September). Pace independent mobile gait biometrics. In 2015 IEEE 7th International Conference on Biometrics Theory, Applications and Systems (BTAS) (pp. 1-8). IEEE.

[2] Zhong, Y., & Deng, Y. (2014, September). Sensor orientation invariant mobile gait biometrics. In IEEE international joint conference on biometrics (pp. 1-8). IEEE.

Response: Thank you very much for your suggestion. To tell you the truth, I also noticed the method of machine vision. We have obtained a lot of public data sets and are currently doing the preliminary work of image processing. Thank you very much for your suggestions, which strengthened my research direction. I hope you can review my next paper on machine vision again next time.

Thank you again for the two papers you provided. I will carefully read and learn their methods and ideas.
